# BglaTNB6, a tailocin produced by a plant-associated nonpathogenic bacterium, prevents rice seed-borne bacterial diseases

Yusuke Kouzai[1]*, Yoshiyuki Sagehashi[1], Riku Watanabe[1], Hideyuki Kajiwara[2], Nobuhiro Suzuki[2], Hiroshi Ono[3], Ken Naito[4], Chiharu Akimoto-Tomiyama[1]*

1 Crop Stress Management Group, Division of Plant Molecular Regulation Research, Institute of Agrobiological Sciences, NARO, Tsukuba, Ibaraki, Japan, 2 Biomacromolecules Research Unit, Research Center for Advanced Analysis, NARO, Tsukuba, Ibaraki, Japan, 3 Bioactive Chemical Analysis Unit, Research Center for Advanced Analysis, NARO, Tsukuba, Ibaraki, Japan, 4 Plant Resources Unit, Research Center of Genetic Resources, NARO, Tsukuba, Ibaraki, Japan

* yusuke.k@affrc.go.jp (YK); akimotoc@affrc.go.jp (CA-T)

**Data Availability Statement:** Genome sequencing of *B. gladioli* strains and the bacterial microbiome analysis have been deposited in the DNA Data Bank

## Abstract

Rice seed-borne diseases caused by the bacterial pathogens *Burkholderia glumae* and *B. plantarii* pose a major threat to rice production worldwide. To manage these diseases in a sustainable manner, a biocontrol strategy is crucial. In this study, we showed that *B. gladioli* NB6 (NB6), a nonpathogenic bacterium, strongly protects rice from infection caused by the above-mentioned pathogens. NB6 was isolated from the indica rice cultivar Nona Bokra seedlings, which possesses genetic resistance to *B. glumae*. We discovered that cell suspensions of NB6 and its culture filtrate suppressed the disease symptoms caused by *B. glumae* and *B. plantarii* in rice seedlings, which indicated that NB6 secretes a plant-protective substance extracellularly. Through purification and mass spectrometry analysis of the culture filtrate, combined with transmission electron microscopy and mutant analysis, the substance was identified as a tailocin and named BglaTNB6. Tailocins are bacteriotoxic multiprotein structures morphologically similar to headless phage tails. BglaTNB6 exhibited antibacterial activity against several *Burkholderia* species, including *B. glumae*, *B. plantarii*, and *B. gladioli*, suggesting it can prevent pathogen infection. Interestingly, BglaTNB6 greatly contributed only to the biocontrol activity of NB6 cell suspensions against *B. plantarii*, and not against *B. glumae*. BglaTNB6 was shown to be encoded by a prophage locus lacking genes for phage head proteins, and a *B. gladioli* strain with the coded BglaTNB6-like locus equipped with phage head proteins failed to prevent rice seedlings from being infected with *B. plantarii*. These results suggested that BglaTNB6 may enhance the competitiveness of NB6 against a specific range of bacteria. Our study also highlights the potential of tailocin-producing endophytes for managing crop bacterial diseases.

## Author summary

Today, there is a strong demand for biopesticides that have a lower environmental impact than chemical pesticides and meet the health and food safety standards expected by

of Japan Sequence Read Archive (https://www.ddbj.nig.ac.jp/dra/index-e.html) under accession numbers DRA017663 (*B. gladioli* NB6), DRA017664 (*B. gladioli* M1064) and DRA017653. All the other data that support the findings of this study are available within the paper and its Supplementary Information. Source data are provided with this paper.

**Funding:** The author(s) received no specific funding for this work.

**Competing interests:** The authors have declared that no competing interests exist.

consumers. Rice bacterial pathogens, particularly *Burkholderia glumae* (Bg) and *B. plantarii* (Bp), are a major concern for rice production worldwide. Here, we describe a novel and promising biocontrol agent, *B. gladioli* NB6 (hereafter NB6), which strongly protects rice against Bg and Bp infection in the susceptible rice cultivar Nipponbare. NB6 was originally found in an indica rice cultivar, Nona Bokra, which was resistant to Bg. Interestingly, bacterial strains of the *Burkholderiaceae* family were identified more frequently in Nona Bokra than in Nipponbare. We showed that cell suspensions of NB6 and its culture filtrate suppressed the disease symptoms caused by Bg and Bp in rice seedlings. Using a mutant that secretes proteins in low-abundance, we then identified a tailocin, named BglaTNB6, that exhibited antagonistic activity against several strains of Bg and Bp but not for the other *Burkholderia* strains. The highly specific antimicrobial activity of tailocins may allow to eliminate pathogens without affecting the beneficial symbiotic bacteria present in agricultural crops. Overall, this study not only reports the discovery of a promising biocontrol agents BglaTNB6 applicable for the management of two rice pathogens, but also suggests that bacterial genetic variants associated with prophage regions may shape bacterial communities through the secretion of tailocins and phages.

## Introduction

Bacterial infections of rice seeds, particularly by *Burkholderia glumae* (Bg) and *Burkholderia plantarii* (Bp), are a major concern for rice production worldwide [1,2]. Rice seeds contaminated with Bg can cause bacterial seedling rot in nursery boxes and bacterial panicle blight in the field, which both lead to serious yield and quality losses [3]. Economic losses associated with the occurrence of these diseases are expected to rise with global warming [4], as relatively high temperatures ranging from 30˚C to 35˚C are conducive to the proliferation of Bg [1]. Bp causes seedling blight in nursery boxes, another serious seed-borne rice disease [5]. The pathogen has been shown to be capable of infecting over 13 genera of Poaceae crops [6]. Both Bg and Bp secrete phytotoxins that serve as key virulence factors [7,8]. Specifically, the former produces an azapteridine toxin known as toxoflavin, while the latter produces a seven-membered nonbenzenoid biotoxin known as tropolone. Both toxins also exhibit antibacterial and antifungal activity against a wide range of microbes [9–12] and can therefore have a negative impact on the agronomic environment.

Controlling the rice seed-borne diseases caused by Bg and Bp is challenging. While this is currently achieved through the use of pesticides, the continuous application of such substances may lead to the development of pesticide-tolerant strains and disturb the agricultural environment [13]. Therefore, investigating alternative approaches for managing these diseases is an imperative issue. Recent research has demonstrated that a mitogen-activated protein kinase kinase kinase (MAPKKK) gene identified in the indica rice cultivar Nona Bokra confers resistance to Bg and Bp [14]. Such genetic resources could pave the way for the development of practical resistant cultivars in the future. Another promising approach involves the use of microbial biocontrol agents. Our previous research has revealed that *Pseudomonas putida* and *Stenotrophomonas maltophilia* strains collected from rice plants suppressed seedling rot symptoms caused by Bg [15]. A *Sphingomonas melonis* strain accumulated and transmitted across generations in disease-resistant rice seeds has been shown to confer resistance against Bp in disease-susceptible cultivars by producing anthranilic acid [6]. Moreover, a *Pantoea dispersa* strain identified from Bg-inoculated rice seeds has been shown to protect rice plants from both

Bg and Bp [12,16]. By using microbial biocontrol methods, it may be possible to manage these pathogens in an environmentally friendly and sustainable manner.

Bacteria often produce proteinaceous toxins known as bacteriocins, which inhibit the growth of similar or closely related bacterial strains [17]. Tailocins, also referred to as phage tail-like particles, are a class of bacteriocins with large molecular weight that are structurally similar to bacteriophage tails [18]. They are protein complexes biosynthesized from prophage-like gene clusters that function by attaching to target bacteria, perforating the cell wall and membrane, and disrupting the proton motive force, which ultimately leads to bacterial death [19]. Unlike pesticides, tailocins have a narrow antibacterial spectrum [18], which makes them potentially valid tools to specifically eliminate pathogens from plants without adversely impacting plant-beneficial bacteria. Indeed, tailocins derived from *Pseudomonas* species have been shown to prevent infection by bacterial pathogens [20,21].

*Burkholderia gladioli* is a plant-associated *Burkholderia* species known primarily as the pathogen responsible for diseases such as gladiolus rot and scab [22]. It also causes grain rot and sheath browning in rice, although it is less prevalent and virulent than Bg [3,23]. However, in spite of its pathogenicity, several nonpathogenic strains of *B. gladioli* have been demonstrated to prevent bacterial and fungal diseases in rice, acting as biocontrol agents [24–27]. In this study, we report on the identification of a tailocin secreted by the newly identified nonpathogenic phyllosphere *B. gladioli* strain NB6, which can strongly protect rice from Bg and Bp.

## Results

### Isolation of *B. gladioli* NB6

It has been shown that several of the seed endophytes isolated from seed possess biocontrol potential [28], and microbes capable of antagonizing a plant pathogen are more likely to be isolated from plants that possess genetic resistance against such pathogen [29]. We therefore focused on bacteria inhabiting the indica rice cultivars Nona Bokra and Bleiyo, which were reported to be resistant to Bg infection [30]. Under our experimental conditions, Nona Bokra was highly resistant to Bg infection (S1A Fig), whereas Bleiyo was not (S1B Fig). We isolated 24 bacterial strains from Nona Bokra, identified by 16S rRNA gene sequencing, and tested their biocontrol activity against Bg using the susceptible rice cultivar Nipponbare. Among these strains, NB6 (identified as *B. gladioli*) effectively suppressed the disease symptoms (Sheet A in S1 Table) and was found to be nonpathogenic to rice (S1C Fig). We observed that bacterial strains belonging to the *Burkholderia* genus were more frequently isolated from Nona Bokra (27–35%, Sheet A in S1 Table) than from Bleiyo (8%, Sheet B in S1 Table) and Nipponbare (5–15%, S2 Fig), and other previously examined rice cultivars such as IR-BB1, TN1 [12]. This observation was consistent with the comparison of bacterial communities in healthy rice plants between Nona Bokra and Nipponbare (S2 Fig), where the bacterial strains belonging to the *Burkholderiaceae* family were identified more frequently in the former cultivar than in the latter.

### NB6 and its culture filtrate protect rice from Bg and Bp

To evaluate the biocontrol activity of NB6, we compared it with that of a previously identified biocontrol strain, *B. gladioli* MAFF301064 (hereafter referred to as M1064), which has been shown to suppress rice grain rot caused by Bg [24]. We tested the biocontrol activity of the two strains against both Bg and Bp through co-inoculation tests. While both the NB6 and M1064 cell suspensions counteracted the inhibitory effect on rice seedling growth caused by Bg (Fig 1A and 1B), only NB6 suppressed the disease symptoms caused by Bp (Fig 1A and 1C).

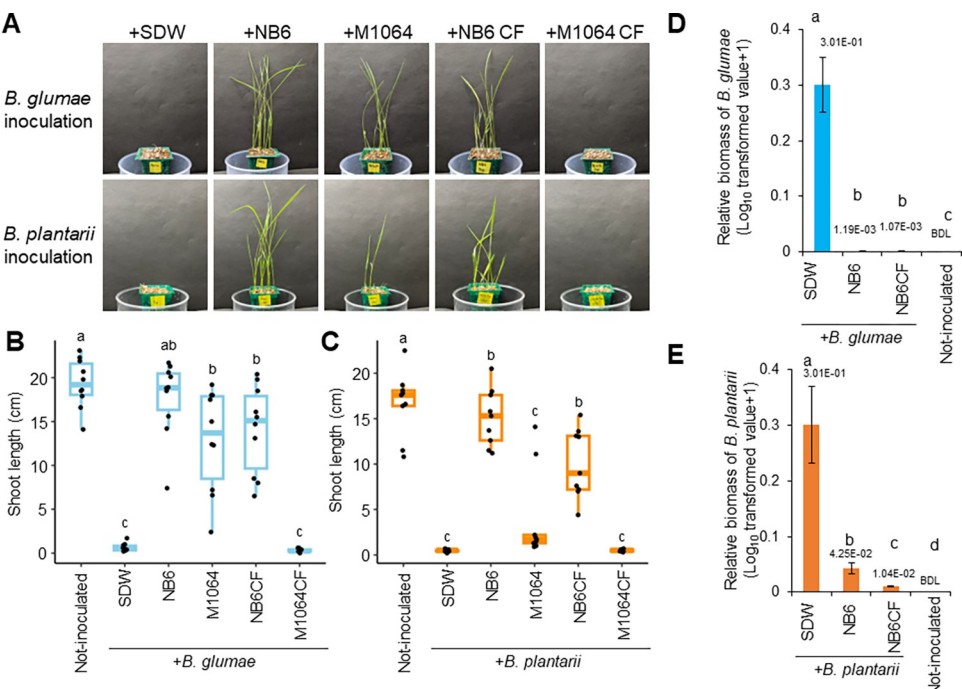

**Fig 1. *B. gladioli* NB6 and its culture filtrate protect rice seedlings from infection by *B. glumae* and *B. plantarii*.**
Rice seeds were co-inoculated with pathogens (Bg: *B. glumae*, Bp: *B. plantarii*) and tested biocontrol bacteria (NB6: *B. gladioli* NB6, M1064: *B. gladioli* MAFF301064), their respective culture filtrates (CF), or sterilized distilled water (SDW). **(A)** Disease symptoms in rice seedlings resulting from the inoculated seeds at 8 days postinoculation (dpi). **(B and C)** Box plots showing the shoot length of rice seedlings at 8 dpi: the central lines indicate the medians, the box limits represent the upper and lower quartiles, and the whiskers extend to the maximum and minimum values. Different letters indicate statistically significant differences between groups (Tukey–Kramer test, P < 0.05, n = 9). The relative biomass of Bg **(D)** and Bp **(E)** in rice seedlings at 8 dpi was measured by quantitative PCR. Two seedlings were used for total DNA extraction. Data are presented as means ± SE of $log_{10}$-transformed values + 1, relative to the plants inoculated with pathogens only. Values are indicated above the bars, and "BDL" signifies below detection limit. Different letters indicate statistically significant differences between groups (Tukey–Kramer test, P <0.05, n = 4). These experiments were repeated three times, and similar results were obtained. Figures show the results of one representative experiment.

To further characterize the biocontrol activity of NB6, we prepared its culture filtrate (CF). Co-inoculation tests demonstrated that the CF of NB6 significantly suppressed both diseases, as observed for its cell suspension, whereas that of M1064 did not (Fig 1A–1C). Moreover, quantitative polymerase chain reaction (qPCR) analyses revealed that both NB6 and its CF significantly reduced pathogen colonization in the inoculated plants (Fig 1D and 1E). These results suggested that NB6 produces unknown plant-protective substance(s) that inhibit pathogen propagation in rice plants, and thus the biocontrol activities of NB6 and M1064 are distinct.

## Identification of genes encoding the plant-protective substance

To identify the plant-protective substance secreted by NB6, we characterized the strain's CF based on its protective activity against Bg. We showed that this CF suppressed disease symptoms and pathogen propagation in a concentration-dependent manner (S3A Fig) and that a short-term heat treatment completely disrupted its protective activity (S3B Fig). Partial purification through ammonium sulfate precipitation revealed that the protective activity of NB6 was concentrated in the 60% saturation fraction (S3C Fig), suggesting that the responsible

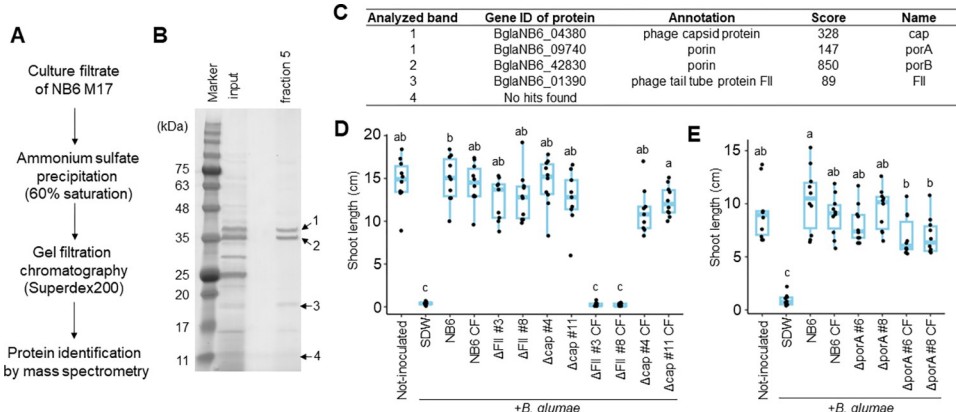

**Fig 2. Purification of the plant-protective substance contained in the culture filtrate of *B. gladioli* NB6. (A)** Schematic representation of the purification and identification processes for proteins contained in the culture filtrate (CF) of *B. gladioli* NB6, which were conducted using a transposon insertion mutant of NB6, NB6-M17. **(B)** Twenty μl of a gel filtration fraction exhibiting protective activity against *B. glumae* (Bg) was analyzed via SDS-PAGE. Arrows indicate four visible bands that were subjected to mass spectrometry analysis. **(C)** Summary of the results of mass spectrometry analysis. **(D and E)** Rice seeds were co-inoculated with Bg and NB6, along with its two independent gene-disruptive mutants (*ΔFII* #3 and #8, *Δcap* #4 and #11, and *ΔporA* #6 and #8), their respective culture filtrates (CFs), or sterilized distilled water (SDW). Box plots show the shoot length of rice seedlings at 7 days postinoculation: the central lines indicate the medians, the box limits represent the upper and lower quartiles, and the whiskers extend to the maximum and minimum values. Different letters indicate statistically significant differences between groups (Tukey–Kramer test, P < 0.05, n = 10). Experiments were repeated twice, and similar results were obtained. Figures show the results of one representative experiment.

substance may be proteinaceous. We next sequenced and assembled the genome of NB6 to identify the bacterial genes encoding this substance. The NB6 genome consisted of two chromosomes and three plasmids, for a total length of 8,890,888 bp and 7,839 annotated protein-coding genes (S2 Table). We further analyzed the CF using a transposon insertion mutant of NB6, NB6-M17, in which the Tn-5 transposon was inserted into the gene encoding a 23S rRNA (adenine2503-C2)-methyltransferase. This mutant exhibited a low diversity and abundance of secreted proteins as well as a slow growth (S4A and S4B Fig) yet, its CF still suppressed the disease symptoms caused by Bg (S4C Fig). We partially purified the CF from NB6-M17 and subjected it to gel filtration chromatography (Fig 2A). Based on the protective activity against Bg, the substance was found to be concentrated in fraction 5 (S5 Fig), which corresponded to the void fraction of the column. This indicated that the substance was likely to be a high-molecular-weight protein or protein complex. The proteins in fraction 5 were separated by sodium dodecyl-sulfate polyacrylamide gel electrophoresis (SDS-PAGE) and four visible bands were analyzed by mass spectrometry (Fig 2B). Although no proteins were detected in band 4, four proteins were identified in the other three bands (Fig 2C), i.e., two porins (designated as porA and porB) and two phage-related proteins (designated as FII and cap).

Next, we generated gene-disruptive mutants of the identified proteins in NB6 through homologous recombination using the pk18mobsacB vector [31,32]. The gene-disruptive mutants for *porA* (BglaNB6_09740), *FII* (BglaNB6_01390), and *cap* (BglaNB6_04380) were successfully created, while that for *porB* (BglaNB6_42830) was not, which suggested that *porB* is essential for NB6 survival. We then prepared the CFs of these mutants and evaluated their protective activity against Bg. While the CFs of *Δcap* and *ΔporA*, similarly to that of NB6, counteracted the inhibitory effect on seedling growth caused by Bg infection, the CF of *ΔFII* displayed an impaired protective activity (Fig 2D and 2E). Moreover, the ectopic expression of

*FII* in *ΔFII* partially complemented the protective activity of the CF of *ΔFII* (S6A Fig). These results indicated that FII is required for the plant-protective activity of the CF of NB6. Intriguingly, the *ΔFII* cell suspension still suppressed the disease symptoms (Fig 2D), suggesting that the disruption of *FII* does not affect the biocontrol activity of NB6 itself against Bg.

## The plant-protective substance is encoded by a prophage-like region

Given that FII was annotated as a phage tail tube protein (Fig 2C) and that the plant-protective substance could be a large molecule (S5 Fig), we hypothesized that the substance might be a phage or a tailocin. To explore this possibility, we predicted prophages within the NB6 genome and identified nine potential candidates (S3 Table). The gene encoding *FII* was located within Region 1, while that encoding cap, another phage-related protein found in the CF of NB6 but not affecting its protective activity (Fig 2C and 2D), was located in Region 2 (Fig 3A). Regions 1 and 2 showed amino acid (aa) identity/similarity with the *Burkholderia* phages phiE202 and AP3, respectively (S3 Table). Although numerous genes exhibited sequence similarities between the two regions, Region 1 lacked genes encoding phage head proteins, unlike Region 2 (Fig 3A). These findings suggested that Region 1 encodes a tailocin that contains *FII*.

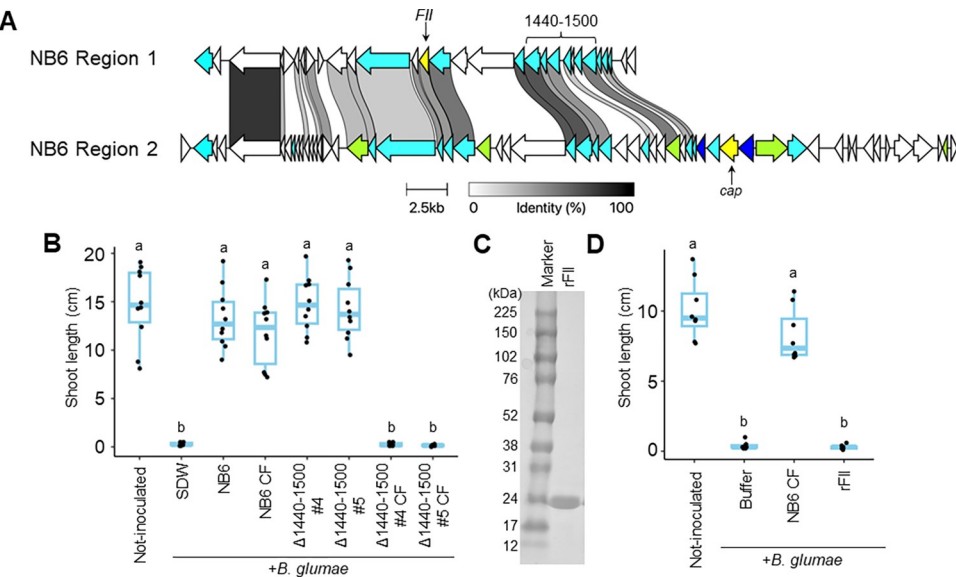

**Fig 3. Identification of a prophage in the *B. gladioli* NB6 genome encoding a plant-protective substance. (A)** Schematic representation of the genetic organization of Region 1 and Region 2 prophages in the *B. gladioli* NB6 genome. Each gene is indicated by arrows with different colors based on their annotations, i.e., white for hypothetical proteins, cyan for phage-related proteins, blue for phage head proteins, and yellow for proteins identified by mass spectrometry. Chords with black–white gradation between regions represent protein similarities. **(B)** Rice seeds were co-inoculated with *B. glumae* (Bg) and NB6, along with its two independent gene-disruptive mutants (*Δ1440–1500* #4 and #5), their respective culture filtrates (CFs), or sterilized distilled water (SDW). The box plot shows the shoot length of rice seedlings at 8 days postinoculation (dpi): the central lines indicate the medians, the box limits represent the upper and lower quartiles, and the whiskers extend to the maximum and minimum values. Different letters indicate statistically significant differences between groups (Tukey–Kramer test, P < 0.05, n = 10). The experiment was repeated twice, and similar results were obtained. **(C)** The recombinant FII protein (rFII) was produced using *Escherichia coli*. The monomeric rFII was purified and resolved via SDS-PAGE. **(D)** Rice seeds were co-inoculated with Bg and the CF of NB6, rFII, or 50 mM Tris-HCl buffer (pH 7.5). The CF and rFII protein concentrations were adjusted to 0.2 mg/ml. The box plot shows the shoot length of rice seedlings at 8 dpi and has the same format as that in **(B)**. Different letters indicate statistically significant differences between groups (Tukey–Kramer test, P < 0.05, n = 8). The experiment was repeated twice, and similar results were obtained. Figures show the results of one representative experiment.

To determine whether the entire Region 1, and not just *FII*, contributed to plant-protective activity, we created another gene-disruptive mutant of NB6, *Δ1440–1500*, which disrupted genes encoding phage-related proteins within Region 1 that are distant from *FII*. The CF of *Δ1440–1500* exhibited an impaired protective activity against Bg (Fig 3B) that was similar to that of *ΔFII* (Fig 2D). Additionally, we produced the recombinant FII protein (rFII) of *Escherichia coli* and purified the monomeric rFII protein using affinity and gel filtration chromatography (Fig 3C). However, rFII could not suppress the disease symptoms (Fig 3D). These results indicated that the FII protein alone does not possess plant-protective activity, and the entire product of Region 1, possibly a tailocin, would be the plant-protective substance.

## *B. gladioli* NB6 produces the antibacterial tailocin BglaTNB6

To identify the product of Region 1, we purified the CF of NB6 using a previously described tailocin/phage purification method [33], which involves ammonium sulfate precipitation and ultracentrifugation. The precipitate obtained by ultracentrifugation, but not the supernatant, was shown to suppress the disease symptoms caused by Bg (Fig 4A), indicating that the plant-protective substance was concentrated in the precipitate. We analyzed this via transmission electron microscopy (TEM) and found that it contained structures recognized as tailocins (Fig 4B), as well as headed phages (Fig 4C–4E). Unlike the headed phages, the tailocins did not exhibit any structural variations, suggesting that NB6 produces a single type of tailocin. The observed substance was estimated to be approximately 145 nm long and 17 nm wide. Tailocin was not observed in the case of *ΔFII* mutant (S6B Fig).

We then used the ultracentrifugation-derived precipitate of NB6 and its gene-disruptive mutants to conduct spot assays. The NB6 precipitate formed inhibition zones on plates with growing Bg and Bp in a concentration-dependent manner but did not form plaques (Fig 4F).

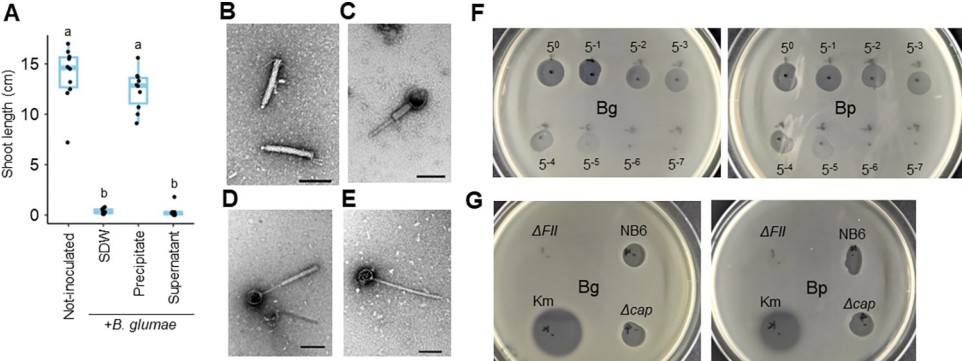

**Fig 4. Identification of the tailocin BglaTNB6 produced by *B. gladioli* with antibacterial activity against *B. glumae* and *B. plantarii*. (A)** Rice seeds were co-inoculated with *B. glumae* (Bg) and either sterilized distilled water (SDW) or the precipitate and supernatant obtained from ultracentrifugation of the culture filtrate of *B. gladioli* NB6. The protein concentrations of the precipitate and supernatant were adjusted to 1.0 mg/ml. The box plot shows the shoot length of rice seedlings at 8 days postinoculation: the central lines indicate the medians, the box limits represent the upper and lower quartiles, and the whiskers extend to the maximum and minimum values. Different letters indicate statistically significant differences between groups (Tukey–Kramer test, P < 0.05, n = 10). The inoculation test was repeated twice, and similar results were obtained. **(B-E)** Structures recognized as tailocins **(B)** and phages **(C-E)** observed in the precipitate of NB6 via transmission electron microscopy. Scale bars indicate 100 nm. Magnification, ×50,000. **(F)** Antibacterial assay of the NB6 precipitate. Five-μl serial dilutions (1/5 stepwise) of the precipitate were spotted onto LB medium containing Bg (Left) and *B. plantarii* (Bp, right) and incubated overnight to evaluate inhibition zones. **(G)** Antibacterial assay of the precipitate of NB6 and its gene-disruptive mutants. Five μl of the precipitate or 0.5 mg/ml kanamycin (Km) were spotted onto LB medium containing Bg (left) and Bp (right) and incubated overnight to evaluate inhibition zones. These antibacterial assays were repeated twice, and similar results were obtained. Figures show the results of one representative experiment.

Similarly, the *Δcap* precipitate formed clear inhibition zones on these plates, whereas the *ΔFII* precipitate did not (Fig 4G). These results indicated that the precipitates of NB6 and *Δcap*, possessed antibacterial activity against Bg and Bp, while that of *ΔFII* did not. Therefore, we concluded that the product of Region 1 was a tailocin, which we designated as BglaTNB6, functioning as a plant-protective substance owing to its ability to suppress Bg and Bp.

We then assessed its antibacterial spectrum using 20 strains of plant-associated *Burkholderia* species (S7A and S7B Fig). The antibacterial activity of BglaTNB6 was effective across strains of Bg, Bp, and *B. gladioli*, except for one strain of Bp. In contrast, strains of *Burkholderia* sp., *B. vietnamiensis* and *Paraburkholderia* sp. were insensitive to BglaTNB6. Interestingly, plaque formation was observed in one strain of *Burkholderia* sp. (S7C Fig), suggesting that one of the phages observed in the NB6 ultracentrifugation precipitate possibly infected it. Therefore, the antibacterial spectrum of BglaTNB6 may be limited to strains of Bg, Bp, and *B. gladioli*. Further detailed experiments to reveal the spectrum of BglaTNB6 are necessary.

## BglaTNB6 underpins the antagonistic activity of NB6

Since the *ΔFII* cell suspension, like NB6, counteracted the inhibitory effects on rice plant growth caused by Bg (Figs 2D and 3B), BglaTNB6 did not seem to be essential for the biocontrol activity of NB6. To evaluate the activity of cell suspension on Bp, we tested the biocontrol activity of *ΔFII* against Bp. The CF of *ΔFII* did not suppress the disease symptoms caused by this pathogen and the *ΔFII* cell suspension also exhibited a significantly reduced protective activity (Fig 5). These results indicated that BglaTNB6 greatly contributed to the biocontrol activity of NB6 against Bp. Similarly, we demonstrated that both M1064 and its CF failed to suppress the disease symptoms caused by Bp (Fig 1A and 1C). We then sequenced the genome of M1064 and predicted two prophages within it (S2 and S3 Tables). We found that Region 1 of NB6 showed sequence similarity to Region 1 of M1064, with both regions being located between the genes encoding cytochrome c and beta-galactosidase (S8 Fig), suggesting a syntenic relationship between the two prophages. However, Region 1 of M1064 appeared to encode a headed phage, not a tailocin, as it contained genes encoding phage head proteins (S8 Fig). This implied that M1064 does not produce BglaTNB6 and thus could not suppress the disease symptoms caused by Bp. In the case of Bp treatment, our findings revealed BglaTNB6 as a key factor for the biocontrol activity of NB6 and the enhancement of the interbacterial competitiveness of NB6 in planta.

## Discussion

In the present study, we demonstrated that NB6, a nonpathogenic strain of *B. gladioli* (S1 Fig), protects rice seedlings against both Bg and Bp infections (Fig 1). Notably, NB6 almost completely restored the severe growth defects caused by these pathogens (Fig 1). We have previously reported several promising biocontrol agents (e.g., *Pseudomonas putida* RSBs and *Pantoea dispersa* BB1) however, these agents provided significant, yet incomplete, recovery [12,15]. Therefore, we speculated that NB6 might have a distinct feature compared to the other agents. Based on our analyses of the properties of NB6, we concluded that the plant-protective substance secreted in the CF of this strain was a tailocin, which we designated as BglaTNB6. To our knowledge, this is the first report of a tailocin produced by a phyllosphere *Burkholderiaceae* microbe. Because this tailocin is a phage variant that is inserted into the bacterial genome (prophage), it is implied that another microbe–microbe interaction via these phages and phage-like particles could shape the phyllosphere microbiota especially at the timing of pathogen invasions.

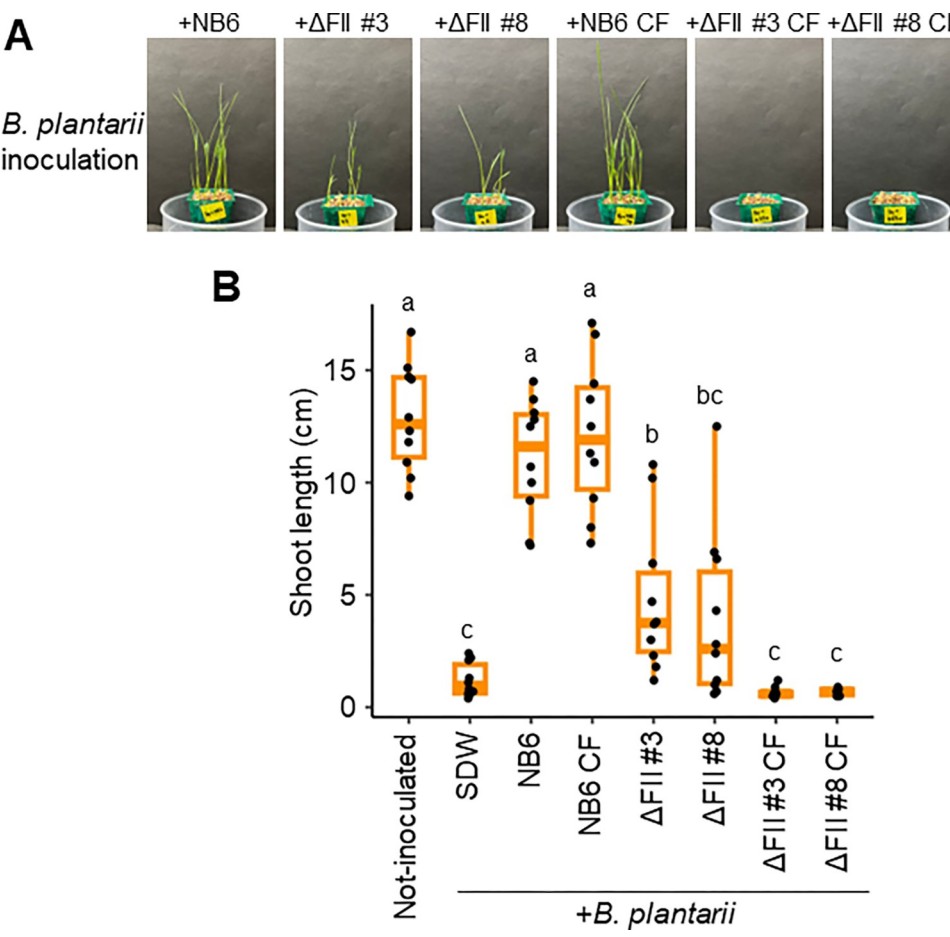

**Fig 5. BglaTNB6 greatly contributed for the biocontrol activity of *B. gladioli* NB6 against *B. plantarii*.** Rice seeds were co-inoculated with *B. plantarii* (Bp) and *B. gladioli* NB6, along with its two independent gene-disruptive mutants (*ΔFII* #3 and #8), their respective culture filtrates (CFs), or sterilized distilled water (SDW). **(A)** Disease symptoms in rice seedlings resulting from the inoculated seeds at 7 days post inoculation (dpi). **(B)** The box plot shows the shoot length of rice seedlings at 7 dpi: the central lines indicate the medians, the box limits represent the upper and lower quartiles, and the whiskers extend to the maximum and minimum values. Different letters indicate statistically significant differences between groups (Tukey–Kramer test, P < 0.05, n = 10). The experiment was repeated three times, and similar results were obtained. Figure shows the results of one representative experiment.

The purified fraction of the CF that exhibited protective activity against Bg contained the FII protein (Fig 2), which was encoded by a gene within the Region 1 prophage (Fig 3). While recombinant FII protein (rFII) itself did not show protective activity, its gene, along with other genes within Region 1, was required for it (Figs 2 and 3). The ultracentrifugation-derived CF precipitate, which contained a single type of tailocin, exhibited protective activity (Fig 4). Moreover, the precipitate demonstrated antibacterial activity against Bg and Bp, with *FII* being essential for it (Fig 4). Therefore, BglaTNB6 would protect rice seedlings from infections by Bg and Bp owing to its antibacterial activity. However, BglaTNB6 was not found to be essential for the biocontrol activity of NB6 against Bg (Figs 2D and 3C), indicating that NB6 employs other antagonistic mechanisms against this pathogen. Interbacterial competition is often mediated by contact-dependent growth inhibition (CDI) [34]. This process relies on cell-to-cell contact and kills neighboring cells by delivering antibacterial compounds using diverse secretion systems [35,36]. Given that multiple *Burkholderia* species also conserve the CDI

system [37–39], it is plausible that the NB6 cell suspension exerts its biocontrol activity against Bg through it. Some examples exist where biocontrol bacteria protect plants from pathogens through multiple mechanisms [40]. For instance, the rhizosphere bacterium *Pseudomonas putida* IsoF isolated from tomato possesses both the ability to directly kill pathogenic bacteria through contact-dependent inhibition (CDI) and to induce systemic resistance in the plant by producing acyl-homoserine lactone (AHL) [36,41]. In contrast, BglaTNB6 was shown to greatly contribute to the biocontrol activity against Bp. Disruption of the *FII* gene in NB6 impaired the protective activity of both the CF and cell suspensions (Fig 5). The previously reported biocontrol strain *B. gladioli* M1064, whose genome did not harbor the prophage encoding BglaTNB6, also failed to suppress the disease symptoms caused by Bp (Figs 1 and S8). Our findings strongly suggested that NB6 secretes BglaTNB6 in rice plants and that the released substance enhances the host's competitiveness against a specific range of bacteria. Additionally, we observed a syntenic relationship between the Region 1 prophage of NB6 and M1064 (S8 Fig). The former was identified as the locus encoding BglaTNB6, while the latter appeared to encode a headed phage. Since tailocin loci are thought to have evolved from prophages [42,43], Region 1 of NB6 might have evolutionarily transitioned from Region 1 of M1064.

NB6 was isolated from the indica rice cultivar Nona Bokra (S1 Table), whose seedlings exhibit the strongest resistance to Bg among 49 rice cultivars [30]. Another previously reported resistant indica cultivar [30], Bleiyo, was shown to be susceptible under our experimental conditions (S1 Fig), which emphasizes the superiority of Nona Bokra. While the resistance of Nona Bokra has been primarily attributed to a MAPKKK gene [14], recent research has suggested that the endophytic microbiota within rice plants also contribute to the resistance to Bg and Bp [6,15]. We observed that species belonging to the *Burkholderiaceae* family were more abundant in the bacterial community of Nona Bokra than in that of the susceptible cultivar Nipponbare (S2 Fig) and Bleiyo (Sheet B in S1 Table), and several *B. gladioli* strains with biocontrol activity against Bg, including NB6, were isolated from Nona Bokra (S1 Table). Therefore, the bacterial community of this cultivar, which contains several beneficial *B. gladioli* strains, may also contribute to its resistance. Furthermore, as the recent discovery of the "M gene" [44] that coded as microbiome-shaping genes in plants, Nona Bokra also might genetically shape the microbiota exclusive to *Burkholderiaceae*.

In the study, we successfully identified BglaTNB6 using NB6-M17, which lacks the 23S rRNA (adenine2503-C2)-methyltransferase that controls translation accuracy in *E. coli* [45]. Despite the limited protein secretion compared to the wild type NB6 (S4A Fig), its CF was shown to be sufficient to protect plants from pathogenic infection (S4C Fig). Therefore, at least the four proteins identified in this study, including two porins and two phage-related proteins, could be translated independently of the enzyme. This knowledge may help to understand the mechanisms regulating the prophage and its excision.

BglaTNB6 exhibited antibacterial activity against strains of Bg, Bp, and *B. gladioli*. Tailocins are generally categorized either into the rigid and contractile-tailed type (R-type) or the flexible and noncontractile tailed type (F-type) [19,46]. Based on the TEM-derived image, BglaTNB6 was classified as an R-type (Fig 4B). Currently, BceTMilo is the only tailocin reportedly found in *Burkholderia* species, also categorized as an R-type [33]. BceTMilo is produced by the human pathogen *B. cepacia* complex and displays antibacterial activity against a wide range of *Burkholderia* species as well as *P. aeruginosa*. Bacterial lipopolysaccharides (LPS) have been shown to function as receptors for tailocins [47], and it has been demonstrated that BceTMilo recognizes the α-D-glucose residue in the LPS of target strains [33]. In contrast to BceTMilo, the antibacterial spectrum of BglaTNB6 (S7 Fig), appears to be narrower which makes it potentially specialized to eliminate only *Burkholderia* species genetically close to *B. gladioli*

[22]. Therefore, the receptors for BceTMilo and BglaTNB6 are likely distinct structures. Given that a Bp strain exhibited tolerance to BglaTNB6 (S7 Fig), elucidating its receptor structures may facilitate the effective use of BglaTNB6 and NB6 to control both Bg and Bp in rice.

The utilization of plant-associated and nonpathogenic bacteria that produce tailocins is a promising strategy for managing bacterial diseases in crops. Other research on these substances has demonstrated that *Allorhizobium vitis* VAR03-1 produces the F-type tailocin Rhizoviticin, which acts as the main biocontrol agent against crown gall disease in grapevines [43]. As both tailocins and phages are not harmful to plants and animals, those specifically targeting bacterial pathogens have been explored for use as biopesticides in crop disease management [48,49]. Currently, these substances are collected from various environments and directly applied to crops or soils. However, this approach often results in incomplete management due to exposure to environmental factors, such as UV irradiation, pH, and soil types, which can compromise their antibacterial activity and viability [50–52]. Therefore, the biocontrol strategy would be significantly improved by protecting tailocins and phages from these environmental factors and delivering them to the pathogen-infected sites through plant-associated bacteria. However, to achieve this, it is crucial to understand the regulatory mechanisms underlying the production of tailocins. Based on its plant-protective activity, NB6 appears to spontaneously produce BglaTNB6 in culture media and plants. Tailocin production is generally induced by the SOS response triggered by cell damage [18]. The production of Rhizoviticin (F-type tailocin) is also induced by the *recA*-dependent SOS response and is repressed by *lexA*, which is located within the Rhizoviticin gene cluster [43]. Elucidating the mechanisms regulating BglaTNB6 production could accelerate the development of practical biocontrol strategies using tailocin-producing bacteria.

## Materials and methods

### Plant and bacterial materials

Rice cultivars Nipponbare (japonica), Nona Bokra (indica), and Bleiyo (indica) were used in this study. Rice seeds were harvested from the experimental paddy fields of the National Agricultural Research Organization (NARO), Tsukuba, Japan and stored in dry boxes at 4˚C until the experiments were performed. Rice plants were grown in sterilized soil (Bonsol No. 2, Sumitomo Chemical, Tokyo, Japan) or 1/2 MS medium containing 0.3% (w/v) Gelrite (FUJI-FILM Wako, Osaka, Japan) in a growth chamber for 14 h under light at 28˚C and for 10 h in the dark at 25˚C. The bacterial strains assigned to MAFF numbers were obtained from the NARO Genebank. *B. gladioli*, *B. plantarii*, *B. glumae*, *Burkholderia* sp., and *Paraburkholderia* sp. were routinely grown in liquid LB medium with shaking (180 rpm) at 30˚C.

### Bacterial microbiome analysis

Surface-sterilized rice seeds of cultivars Nipponbare and Nona Bokra were grown in soil for 7 days. The resulting plant shoots were collected for DNA extraction, which was carried out using the DNeasy Plant Mini Kit (Qiagen, Hilden, Germany). The library preparation, sequencing, and preprocessing of the obtained reads for 16S rRNA amplicon sequencing were performed by the Bioengineering Lab. Co., Ltd. (Sagamihara, Japan). Sequencing libraries were constructed using the 2-step tailed PCR method with primers targeting the V3-V4 region of the bacterial 16S rRNA gene. PCR reactions were supplemented with peptide nucleic acid to eliminate amplification from plant mitochondria and chloroplasts [53]. Libraries were sequenced using a Miseq sequencer (Illumina, San Diego, CA, USA). The obtained reads were preprocessed using the FASTX-Toolkit (v 0.0.14) and sickle tool (v 1.3) to remove reads with quality scores < 20 and lengths < 130 bases, and paired-end reads were merged using FLASH

(v 1.2.11). The preprocessed reads were subjected to microbiome analysis using the QIIME2 pipeline (v. 2023.5) [54]. After removing noise and chimeric sequences using the DADA2 plugin, taxonomy assignment was performed at 97% similarity using the feature classifier plugin with the Silva 138 database as a reference [55]. Features with a frequency lower than 10 or that were assigned to plant chloroplasts and mitochondria were excluded from the analysis. The datasets were submitted to the DNA Data Bank of Japan Sequence Read Archive (https://www.ddbj.nig.ac.jp/dra/index-e.html) under accession number DRA017653.

## Isolation of *B. gladioli* NB6 from rice

*B. gladioli* NB6 was isolated from Nona Bokra rice plants inoculated with or without Bg using the previously described method [12]. The isolated bacteria from Nona Bokra and Bleiyo were identified through 16S rRNA gene sequencing using the primers listed in S4 Table. Those 16S rRNA gene sequences were submitted to the DNA Data Bank of Japan Sequence Read Archive (https://www.ddbj.nig.ac.jp/dra/index-e.html) under accession numbers LC831823 to LC831846.

## Inoculation test

The bacterial strains (Bg and Bp) were cultured in liquid LB medium with shaking at 30˚C until the OD600 exceeded 2.0. Bacterial cells were collected by centrifugation, washed twice, and resuspended in SDW. Inocula were prepared by mixing the Bg or Bp cell suspension (OD600 of 0.008) with an equal volume of tested bacterial cell suspension (OD600 of 0.008), their CF, or tailocin suspensions. Surface-sterilized rice seeds (cv. Nipponbare) were vacuum-inoculated for 5 min, then gently mixed for 1 hour. After removing the excess bacterial suspension from the seeds, incubated on sterilized soil or a 1/2 MS plate for 7–8 days. The colony-forming unit (CFU) of the excess bacterial suspension was measured and confirmed for viability (S9 Fig). Disease severity was evaluated by measuring shoot length and pathogen propagation in the inoculated plants, as previously described [6,12,16]. The primers used for measuring pathogen propagation are listed in S4 Table.

## CF preparation

Bacterial strains were cultured in liquid LB medium with shaking at 30˚C until the OD600 exceeded 2.0. The CFs were prepared by removing the bacterial cells through centrifugation (8,000 rpm, 5 min, 20˚C) and subsequent filtration using a 0.22-μm-pore filter (Thermo Fisher Scientific, Waltham, MA).

## Identification of proteins in the CFs

The proteins contained in 100 ml of the CF of *B. gladioli* NB6 were precipitated in 60% saturated ammonium sulfate for 1 hour on ice. They were then collected by centrifugation (10,000× g, 20 min, 4˚C), dissolved in 2.5 ml of 50 mM Tris-HCl buffer (pH 7.5), desalted using a PD-10 column (Cytiva, Marlborough, MA), and eluted in 4 ml of the same buffer. Subsequently, 0.5 ml of the eluate was subjected to a Superdex 200 Increase 10/300 GL gel filtration column (Cytiva). The proteins were eluted in the same buffer and fractionated every 2 ml. Each fraction was concentrated 5 times using a VivaSpin ultrafiltration column (MWCO 5,000, Sartorius) and then subjected to SDS-PAGE and disease resistance tests. The proteins in the fractions that exhibited plant-protective activity were identified using mass spectrometry analysis. The proteins in SDS-PAGE gel slices were digested using the Trypsin Profile IGD Kit (Sigma-Aldrich, St. Louis, MO), according to the manufacturer's instructions, and subjected

to MALDI TOF/TOF MS analysis using a 4800plus MALDI TOF/TOF Analyzer (Applied Biosystems, Foster City, CA). The obtained data were searched against the *B. gladioli* NB6 protein database based on the sequenced genome using the MASCOT server (version 2.5).

## Genome sequencing of *B. gladioli* strains

Bacterial gDNA was extracted from *B. gladioli* NB6 and M1064 cells using the NucleoBond DNA HMW kit (TaKaRa, Shiga, Japan). For the NB6 genome analysis, a sequencing library was prepared using the SMRTbell Express Template Prep Kit (PacBio, Menlo Park, CA) and sequenced using a PacBio Sequel sequencer (PacBio). These procedures were performed by Macrogen Inc. (Seoul, South Korea). The obtained reads were filtered to exclude those <5 kb in length and assembled using Flye (v2.8) [56]. For the M1064 genome, a sequencing library was prepared using the Ligation Sequencing Kit (Oxford Nanopore Technologies, Oxford, UK) following the manufacturer's instructions, and sequenced using a PromethION sequencer (Oxford Nanopore). The obtained reads were similarly filtered and assembled using the Flye-based assembler, Dragonflye (v1.0.5). Both assembled genomes were annotated using the DDBJ Fast Annotation and Submission Tool. The datasets were submitted to the DNA Data Bank of Japan Sequence Read Archive (https://www.ddbj.nig. ac.jp/dra/index-e.html) under accession numbers DRA017663 (*B. gladioli* NB6) and DRA017664 (*B. gladioli* M1064).

## Prediction of prophages

Prophages were identified using the web-based phage search tools PHAST and PHASTEST [57,58] with manual annotation. Visualization and comparison of the predicted prophages were performed using Clinker [59].

## Creation of *B. gladioli* NB6 mutants

The preparation of competent *B. gladioli* cells and their transformation were carried out using a previously described method [60]. NB6-M17, a mutant characterized by the secretion of proteins with low diversity and abundance, was generated through transposon-mediated random mutagenesis using the EZ-Tn<KAN-2>Tnp Transposome Kit (Epicentre Biotechnologies, Madison, WI), following the manufacturer's instructions. The gene inserted by the Tn5 transposon was identified through inverse PCR and Sanger sequencing, using the primers provided in the kit (S4 Table). Gene deletions were performed using the homologous-recombination-based method with pK18mobsacB, as previously described [33,61]. The upstream and downstream regions of the targeted genes (400 bp) were artificially synthesized and tandemly cloned into pK18mobsacB. This process was performed using GenScript (Piscataway, NJ). Plasmid constructs were transformed into NB6 cells by electroporation and selected on LB agar containing kanamycin. The second homologous recombination event was induced on LB agar containing 20% sucrose. The deletion of the targeted genes was verified by PCR and Sanger sequencing using the primers listed in S4 Table. Complementation of *ΔFII* was performed via the ectopic expression of FII using the broad-host-range vector pHM1, as previously described [62,63]. The *FII* coding sequence was cloned into pHM1 and sequenced using the primers listed in S4 Table. The plasmid construct was transformed into *ΔFII* cells and selected on LB agar containing spectinomycin. By using two independent mutant strains in the experiments, we confirmed that the observed phenotype was not due to a spurious mutation arising during the mutant generation process.

## Production of the recombinant FII protein

The codon-optimized coding sequence of *FII* was artificially synthesized and cloned into pGEX-6P-1 using GenScript. The plasmid construct was expressed in *E. coli* BL21(DE3) by 0.1-mM IPTG at 18˚C with overnight shaking. The recombinant FII protein was extracted from *E. coli* cells and subjected to a GST affinity column (Glutathione Sepharose 4B, Cytiva). After on-column protease digestion with PreScission Protease for GST-tag removal (Cytiva), the eluate was further purified using a Superdex75 10/300 GL gel filtration column (Cytiva).

## Purification and visualization of BglaTNB6

The CF of *B. gladioli* NB6 was partially purified via ammonium sulfate precipitation as described above. The eluate from the desalting column was passed through a 0.22-μm-pore syringe filter (Thermo Fisher Scientific) and subjected to ultracentrifugation (41,000 rpm, 4˚C, 2.5 h). The obtained precipitate was resuspended in 0.1 ml of 50 mM Tris-HCl buffer (pH 7.5). The tailocin suspension was negatively stained with 2% uranyl acetate and visualized using a transmission electron microscope (JEM-2010, JEOL Ltd., Tokyo, Japan) at the Hanaichi Ultra-structure Research Institute (Okazaki, Japan).

## Spot assay

The antibacterial activity of the tailocin suspension was evaluated using a spot assay [33]. Briefly, the suspension was serially diluted with 50 mM Tris-HCl buffer (pH 7.5), and 5 μl of each dilution was spotted onto an LB agar plate overlaid with 5 ml of LB soft agar containing 200 μl of an indicator bacterial cell suspension ($OD_{600}$ = 0.2). The inhibition zones on the plates were assessed after overnight incubation at 30˚C. The reciprocal of the highest dilution that formed a visible inhibition zone was defined as the value of the relative activity, which was expressed in arbitrary units.

## Supporting information

**S1 Fig. Pathogenicity of *B. glumae* and *B. gladioli* NB6 to rice. (A)** Rice seeds (Nona Bokra) were inoculated with *B. glumae* or sterilized distilled water (SDW). Photos were taken at 7 days postinoculation (dpi). **(B)** Rice seeds (Bleiyo) were inoculated with *B. glumae* or sterilized distilled water (SDW). Photos were taken at 7 dpi. **(C)** Rice seeds (Nipponbare) were inoculated with *B. gladioli* NB6 or sterilized distilled water (SDW). Photos were taken at 8 dpi. The box plot shows the shoot length of rice seedlings at 8 dpi: the center lines indicate the medians, the box limits represent the upper and lower quartiles, and the whiskers extend to the maximum and minimum values. Different letters indicate statistically significant differences between groups (Tukey–Kramer test, P < 0.05, n = 10). These experiments were repeated twice, and similar results were obtained. Figures show the results of one representative experiment.
(TIF)

**S2 Fig. Comparison of rice bacterial communities between Nipponbare and Nona Bokra.** Relative abundance of bacterial families in rice plants of the cultivars Nipponbare and Nona Bokra.
(TIF)

**S3 Fig. Characterization of the culture filtrate of B. gladioli NB6. (A)** Rice seeds were coinoculated with *B. glumae* (Bg) and *B. gladioli* NB6 or its culture filtrate (CF) diluted with liquid LB medium. Upper: the box plot shows the shoot length of rice seedlings at 7 days

postinoculation (dpi): the center lines indicate the medians, the box limits represent the upper and lower quartiles, and the whiskers extend to the maximum and minimum values. Different letters indicate statistically significant differences between groups (Tukey–Kramer test, $P <$ 0.05, n = 8). Lower: the bar chart represents the relative biomass of Bg in rice seedlings at 7 dpi, measured by quantitative PCR. Two seedlings were bulked for total DNA extraction. Data are presented as means ± SE of values relative to plants inoculated only with the pathogens. Different letters indicate statistically significant differences between groups (Tukey–Kramer test, $P < 0.05$, n = 4). **(B)** Rice seeds were co-inoculated with Bg and NB6, its CF, the CF treated with 95˚C for 5 minutes, or sterilized distilled water (SDW). Upper: the box plot shows the shoot length of rice seedlings at 7 dpi, following the same format as in (A). Different letters indicate statistically significant differences between groups (Tukey–Kramer test, $P < 0.05$, n = 10). Lower: the bar chart represents the relative biomass of Bg in rice seedlings at 7 dpi, following the same format as in (A). Different letters indicate statistically significant differences between groups (Tukey–Kramer test, $P < 0.05$, n = 4). **(C)** Rice seeds were co-inoculated with Bg and the fractions of ammonium sulfate precipitation of the CF of NB6 or 50 mM Tris-HCl buffer (pH 7.5). The box plot shows the shoot length of rice seedlings at 7 dpi, following the same format as in (A). Different letters indicate statistically significant differences between groups (Tukey–Kramer test, $P < 0.05$, n = 9). These experiments were repeated two times, and similar results were obtained. Figures show the results of one representative experiment. (TIF)

**S4 Fig. Characterization of the mutant of *B. gladioli* NB6, NB6-M17. (A)** Twenty μl of the 60% saturation fractions obtained from ammonium sulfate precipitation of the culture filtrates (CF) of *B. gladioli* NB6 and NB6-M17 were resolved by SDS-PAGE. **(B)** Initial growth of NB6 and NB6-M17 cultured in liquid LB medium. Bacterial growth was measured as $OD_{600}$ every hour. Error bars represent SE (n = 4). The experiment was repeated twice with similar results obtained. **(C)** Rice seeds were co-inoculated with *B. glumae* (Bg) and the CF of NB6, the CF of NB6-M17, or sterilized distilled water (SDW). The box plot shows the shoot length of rice seedlings at 7 days postinoculation: the center lines indicate the medians, the box limits represent the upper and lower quartiles, and the whiskers extend to the maximum and minimum values. Different letters indicate statistically significant differences between groups (Tukey–Kramer test, $P < 0.05$, n = 7). These experiments were repeated two times, and similar results were obtained. Figures show the results of one representative experiment. (TIF)

**S5 Fig. Purification of the culture filtrate of NB6-M17. (A)** Gel filtration chromatogram of the 60% saturation fractions obtained from ammonium sulfate precipitation of the culture filtrate (CF) of *B. gladioli* NB6-M17. **(B)** Rice seeds were co-inoculated with *B. glumae* (Bg) and the fractions obtained from the gel filtration chromatography. Three fractions were bulked for inoculation tests. The box plot shows the shoot length of rice seedlings at 7 days postinoculation (dpi): the center lines indicate the medians, the box limits represent the upper and lower quartiles, and the whiskers extend to the maximum and minimum values. Different letters indicate statistically significant differences between groups (Tukey–Kramer test, $P < 0.05$, n = 7). **(C)** Rice seeds were co-inoculated with Bg and the fractions obtained from the gel filtration chromatography. The box plot shows the shoot length of rice seedlings at 7 dpi, following the same format as in (B). Different letters indicate statistically significant differences between groups (Tukey–Kramer test, $P < 0.05$, n = 7). These experiments were repeated two times, and similar results were obtained. Figures show the results of one representative experiment. (TIF)

**S6 Fig. Complementation analysis of *ΔFII*.** (A) *FII* was ectopically expressed using the pHM1 vector in *ΔFII* (Comp). Rice seeds were co-inoculated with *B. glumae* and *B. gladioli* NB6, *ΔFII*, Comp, or their respective culture filtrates. The box plot shows the shoot length of rice seedlings at 7 days postinoculation (dpi): the center lines indicate the medians, the box limits represent the upper and lower quartiles, and the whiskers extend to the maximum and minimum values. Different letters indicate statistically significant differences between groups (Tukey–Kramer test, $P < 0.05$, n = 8). These experiments were repeated two times, and similar results were obtained. Figure show the results of one representative experiment. (B) Picture observed in the precipitate of *ΔFII* via transmission electron microscopy. Scale bars indicate 100 nm. Magnification, ×50,000.
(TIF)

**S7 Fig. Antibacterial spectrum of BglaTNB6.** BglaTNB6 suspensions, obtained from the ultracentrifugation of the culture filtrate of *B. gladioli* NB6, were subjected to spot assays. Five μl of the suspensions or 0.5 mg/ml kanamycin were spotted onto LB medium containing the indicator bacterium and incubated overnight to evaluate inhibition zones. The experiment was repeated twice with similar results obtained. **(A)** Photos of inhibition zones formed on the LB media growing indicator bacteria. **(B)** Summary of the antibacterial spectrum. **(C)** Photo of plaques formed on the LB medium growing Burkholderia sp. MAFF301584.
(TIF)

**S8 Fig. Comparison of prophages in the genomes of *B. gladioli* NB6 and *B. gladioli* M1064.** Schematic representation of the genetic organization of Region 1 prophages in the *B. gladioli* NB6 and M1064 genomes. Each gene is indicated by arrows, with colors based on their annotations: white for hypothetical proteins, cyan for phage-related proteins, blue for phage head proteins, and yellow for proteins identified by mass spectrometry. Chords with black-white gradation between Regions represent protein similarities.
(TIF)

**S9 Fig. Viability of the inocula.** The colony-forming unit (CFU) of the excess bacterial suspension of Bg (A) and Bp (B) inocula on LB medium were measured.
(TIF)

**S1 Table. Bacterial strains isolated from Nona Bokra and Bleiyo.**
(XLSX)

**S2 Table. Summary of sequenced genomes in this study.**
(XLSX)

**S3 Table. Summary of predicted prophages.**
(XLSX)

**S4 Table. Primers and oligo DNAs used in this study.**
(XLSX)

**S1 Data. Source data for Figs 1B, 1C, 1D, 1E, 3B, 3D, 4A, 5B, S1, S3A, S3B, S3C, S4C, S5A, S5B, S5C, S6 and S9.**
(ZIP)

# Acknowledgments

We thank Marie Uemura, Sachie Watanabe, and Tomoko Sato for technical assistance. We also thank the NARO Genebank for providing bacterial strains.

## Author Contributions

**Conceptualization:** Yusuke Kouzai, Chiharu Akimoto-Tomiyama.

**Data curation:** Yusuke Kouzai, Riku Watanabe, Chiharu Akimoto-Tomiyama.

**Investigation:** Yusuke Kouzai, Riku Watanabe, Hideyuki Kajiwara, Hiroshi Ono, Ken Naito, Chiharu Akimoto-Tomiyama.

**Resources:** Yoshiyuki Sagehashi, Hideyuki Kajiwara, Nobuhiro Suzuki, Hiroshi Ono, Ken Naito, Chiharu Akimoto-Tomiyama.

**Supervision:** Chiharu Akimoto-Tomiyama.

**Validation:** Yoshiyuki Sagehashi.

**Visualization:** Yusuke Kouzai, Chiharu Akimoto-Tomiyama.

**Writing – original draft:** Yusuke Kouzai, Riku Watanabe, Hideyuki Kajiwara, Nobuhiro Suzuki, Hiroshi Ono, Ken Naito.

**Writing – review & editing:** Yusuke Kouzai, Yoshiyuki Sagehashi, Riku Watanabe, Chiharu Akimoto-Tomiyama.

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
