## [Decision Letter · Decision Letter 0]

15 Jul 2024

Dear Dr Akimoto-Tomiyama,

Thank you very much for submitting your manuscript "BglaTNB6, a tailocin produced by a plant-associated nonpathogenic bacterium, prevents rice seed-borne bacterial diseases" for consideration at PLOS Pathogens. As with all papers reviewed by the journal, your manuscript was reviewed by members of the editorial board and by several independent reviewers. The reviewers appreciated the attention to an important topic. Based on the reviews, we are likely to accept this manuscript for publication, providing that you modify the manuscript according to the review recommendations.

Three reviewers have evaluated the manuscript and all find the experimental data and results clear and compelling. Each reviewer has suggested a variety of editorial changes that will enhance the clarity and readability of the manuscript, but I don't think that new sets of data or experiments are necessary for this submission.

Sincerely,

David A Baltrus

Guest Editor

PLOS Pathogens

Bart Thomma

Section Editor

PLOS Pathogens

Michael Malim

Editor-in-Chief

PLOS Pathogens

orcid.org/0000-0002-7699-2064

Three reviewers have evaluated the manuscript and all find the experimental data and results clear and compelling. Each reviewer has suggested a variety of editorial changes that will enhance the clarity and readability of the manuscript, but I don't think that new sets of data or experiments are necessary for this submission.

Reviewer Comments (if any, and for reference):

Reviewer's Responses to Questions

**Part I - Summary**

Reviewer #1: The authors have demonstrated convincingly that B. gladioli strain NB6 releases a tailocin able to inhibit the growth of strains of B glumae and B plantarii.

Reviewer #2: This manuscript reports that a nonpathogenic stain of Burkholderia gladioli, NB6, shows a strong disease suppression activity against seedling blight caused by B. glumae and B. plantarii, and that a new tailocin (a phage-tail-like bacteriocin) produced by B. gladioli NB6, BglaTNB6 (B6 hereafter), is a key antibacterial element of this antagonistic strain.

Overall, this study was performed very well from the characterization of disease suppression activity by the cell suspension and the cell-free filtrate of B. gladioli NB6 to the identification of the tailocin B6 and demonstration of its direct antibacterial activity against strains of Burkholderia spp. including B. glumae and B. plantarii. Manuscript writing and data presentation were also nicely done, describing each step of molecular genetics and biochemistry experiments very well. I think this study brings some useful information about an important mechanism of biological control against bacterial pathogens.

Reviewer #3: This manuscript describes the isolation and characterization of a B. gladioli strain (NB6), which was isolated from a genetically resistant line of rice (Nona Bokra) and is able to suppress multiple Burkholderia pathogens of rice. In general this paper does a thorough job of establishing the link between the tailocin of NB6 and the disease suppressing capability of this strain. This paper makes an important contribution to the field, however, I have a couple of concerns about the manuscript in its current form.

**Part II – Major Issues: Key Experiments Required for Acceptance**

Reviewer #1: Although not the main focus of the paper they have also alluded to the host genetic control and recruitment of appropriate Burkholderia biocontrol strains as mediating Nona Bokra’s tolerance to these rice pathogens. On this point the evidence is both circumstantial and unconvincing. The variation in bacterial diversity in different cultivars is interesting although there are more changes than just Burkholderia (which counters claims on Line 396) and this also ignores that these are japonica vs indica cultivars. It would be helpful to also look at the Bleiyo microbiome. Based on the authors hypothesis, Bleiyo should lack enrichment for Burkholderia. Also, while testing in rice seedlings is interesting, the authors should conduct in vitro tests for Bg and Bp viability after being mixed with NB6, CF and tailocin using the same conditions for prepping inoculum. What is the amount of time between mixing and seed vacuum infiltration? I'm curious about the M17 transposon mutant. Were you actively seeking mutants with reduced protein secretion? EM analysis should also be conducted of the DFII and DFII complement lines for loss and restoration of tailocin-like structures. Also while BglaTNB6 makes strong contributions to restricting B. plantarii growth it is clearly not “essential”.

Reviewer #2: No major issues found

Reviewer #3: -In general, I found the manuscript reasonably easy to read, however there were a number of sentences in the discussion that I found hard to interpret (I’ve indicated each in minor comments). In my opinion, these sentences need to be addressed.

-Although the authors acknowledge the data showing that there are at least two systems operating to suppress disease (one contact-independent tailocin and another, likely contact dependent system), this is not really addressed at all in the discussion. I think this is a more important point to discuss its implications for disease suppression and if there are other examples in plants (or elsewhere), where a biocontrol strain has multiple systems that contribute to biocontrol.

-There should be more explicit descriptions of how genetic data is confirmed. It looks as though multiple deletion mutants were used in Figs 2, 3, and 5 as a way to confirm that the phenotypes weren’t related to auxiliary mutations? However, I don’t believe this is stated clearly in the text (either in methods or results section). Additionally, the methods section contains language indicating that a complement strain of the FII mutant was generated. I don’t recall seeing this strain in the results (but I might have missed it)?

-L236 – This statement indicates that the capsid proteins are not present in the predicted prophage regions, however, table S3 indicates that this progaphe region was scored as intact, which I would think would be odd if the capsid genes are missing. This discrepancy should be addressed.

**Part III – Minor Issues: Editorial and Data Presentation Modifications**

Reviewer #1: L63: Using a mutant that secretes proteins in low abundance

L111: why would high specificity limit emergence of tolerance in the targeted organism? There are clear examples of mutations that allow evasion of tailocins

L114: citation 19 "The sensitivity of phages to environmental stressors such as UV irradiation and soil (type, pH, moisture, organic matter content) is still an obstacle to phage therapy on plants" Tailocins lack nucleic acid and so should be less sensitive to UV

L128: I see no mention of this pathogen resistance correlating with biocontrol strains in Truyens et al The Upreti and Thomas seems very limited and circumstantial.

L134: Upload the 16S sequences to a public repository

Fig 1D: Relative bacterial population data should be log transformed. Which strains of Bg and Bp were used in inoculation tests?

L205: inhibitory effect on seedling growth

L227: encoded by a prophage-like region

L347: There’s a noticeable shift in writing quality in the first paragraph of the discussion (for instance, distinct feather). Also, why is it unexpected to have a tailocin region in the genome. Where else would it be?

L478: Which strains of Bg and Bp were used in inoculation tests?

Reviewer #2: 1. To distinguish from B. plantarii, ‘Bp’ indicated for base pairs should be ‘bp’ (e.g. lines 185 and 539)

2. Line 269-315: The authors presented a TEM image of the tailocin from the CF fraction of B. gladioli NB6. To claim that the phage tail-like structure is the tailocin B6, the authors should mention or present data that no such structure was observed in the CF of the FII or region1-deleted mutant.

3. It is not clear why the authors mentioned ‘as expected’ in the sentence on line 322. How did they expect that the cell suspension of the FII-deficient mutant would also exhibit reduced protective activity unlike Bg, with the observation that B6 was effective to both Bg and Bp?

4. Line 334: The sentence is for Bp, not for Bg because B6 is not required for the activity of the NB6 cells. The sentence should be modified, reflecting this fact.

5. Please refine the sentence on lines 350 – 353.

6. Line 396: Please provide more information of the ‘M gene’ with one or two sentences.

7. Line 482: Please describe more detail about the vacuum inoculation method, such as duration of vacuum, psi, drying process before planting, etc.

8. Lines 541 and 547: Please describe how the bacterial cells were transformed. Heat shock, electroporation, or conjugation?

Reviewer #3: L43-44 - This sentence should be reworded since species (not strains) are listed.

L49-50 - I don’t think this sentence logically flows form the preceeding sentences, since it is not clear/established that disease suppression is connected to the competitive fitness of NB6.

L60 – Nona Bokra was described as resistant in the abstract, but as tolerant in this line. These (and all other designations in the manuscript) should be harmonized.

L91-94 – This sentence seems to go in a direction not relevant to this manuscript, I think this can be removed.

L135 – Was this strain only tested agains Nipponbare? Or was it tested for pathogenicity toward other rice cultivars?

L135 – 137 - This statement seems to require the reader to compare the number of isolates from table S1 to ref. 12, which makes it unnecessarily complicated to review this statement. This should be more supported directly in this text.

L181 – 182 – How does the 60% saturation fraction confirm proteinaceous identity? Can the authors cite something to support this statement?

L226 – For this line and all similar lines related to the other figures, are the results of one representative experiment shown or are the graphs representative of all of the combined data?

L234 – Please specify what is meant by “homologies” – nt or aa identity/similarity?

L241 – What genes specifically were deleted? Do these numbers refer to the locus tags? Or some other gene identifier?

L250 – Why compare region 1 and 2 in this manner? Do the authors think these two regions are evolutionarily related?

L293-294 – This sentence seems unwarranted given the data. It might be true, but I don’t think there is sufficient data to conclude this.

L322 – itals deltaFII

L336 – This figure heading should be revised, since the data shows that the BglaTNB6 contributes to biocontrol activity (but is not essential, given the other contact dependent system)

L352 – I don’t understand this sentence

L353 – I don’t understand what is meant by “distinct feather”.

L357-358 – This sentence doesn’t make sense to me since tailocins are by definition phage genomes that have inserted into the bacterial genome. What makes this “unexpectedly”?

L358-359 – I don’t get the logic of this clause.

L361-363 – What is “it” in this sentence?

L383 – What tailocin loci don’t originate from prophages that would necessitate “some” for this sentence.

L386-388 – This sentence seems out of place for what is being reported, I suggest removing.

L414 – How does this figure confirm this statement?

L435 – It should be indicated that Rhizoviticin is a tailocin.

PLOS authors have the option to publish the peer review history of their article (what does this mean?). If published, this will include your full peer review and any attached files.

Reviewer #1: No

Reviewer #2: **Yes: **Jong Hyun Ham

Reviewer #3: No

Figure Files:

Data Requirements:

Reproducibility:

References:

---

## [Decision Letter · Decision Letter 1]

23 Sep 2024

Dear Dr Akimoto-Tomiyama,

Thank you very much for submitting your manuscript "BglaTNB6, a tailocin produced by a plant-associated nonpathogenic bacterium, prevents rice seed-borne bacterial diseases" for consideration at PLOS Pathogens. As with all papers reviewed by the journal, your manuscript was reviewed by members of the editorial board and by several independent reviewers. The reviewers appreciated the attention to an important topic. Based on the reviews, we are likely to accept this manuscript for publication, providing that you modify the manuscript according to the review recommendations.

Thank you for your resubmission. All reviewers now agree that the revised manuscript is nearly acceptable for publication. I would just like to see a bit more clear and direct language concerning construction of multiple mutants incorporated into the manuscript as the reviewer suggests.

Sincerely,

David A Baltrus

Guest Editor

PLOS Pathogens

Bart Thomma

Section Editor

PLOS Pathogens

Michael Malim

Editor-in-Chief

PLOS Pathogens

orcid.org/0000-0002-7699-2064

Thank you for your resubmission. All reviewers now agree that the revised manuscript is nearly acceptable for publication. I would just like to see a bit more clearl and direct language concerning construction of multiple mutants incorporated into the manuscript as the reviewer suggests.

Reviewer Comments (if any, and for reference):

Reviewer's Responses to Questions

**Part I - Summary**

Reviewer #1: My concerns have been addressed

Reviewer #3: This paper makes an important contribution to the field of biological controls, particularly tailocin-mediated biological control of an important bacterial rice pathogen. The authors have done a reasonably good job of addressing my previous comments.

**Part II – Major Issues: Key Experiments Required for Acceptance**

Reviewer #1: (No Response)

Reviewer #3: None

**Part III – Minor Issues: Editorial and Data Presentation Modifications**

Reviewer #1: (No Response)

Reviewer #3: It would still be good for the authors to state explicitly in the text (either figure legends, results section, or methods section) that the multiple mutant isolates shown in figures 2, 3, and 5 are all included to confirm that the phenotype of the mutants is related to the deletion that was introduced, and not a spurious mutation that was picked up during the mutant generation process. I think this is why multiple mutant isolates are included in the figures, but it isn't actually confirmed in any text.

PLOS authors have the option to publish the peer review history of their article (what does this mean?). If published, this will include your full peer review and any attached files.

Reviewer #1: No

Reviewer #3: **Yes: **Kevin L. Hockett

Figure Files:

Data Requirements:

Reproducibility:

References:

---

## [Editor Report · Decision Letter 2]

4 Oct 2024

Dear Dr Akimoto-Tomiyama,

We are pleased to inform you that your manuscript 'BglaTNB6, a tailocin produced by a plant-associated nonpathogenic bacterium, prevents rice seed-borne bacterial diseases' has been provisionally accepted for publication in PLOS Pathogens.

Best regards,

David A Baltrus

Guest Editor

PLOS Pathogens

Bart Thomma

Section Editor

PLOS Pathogens

Michael Malim

Editor-in-Chief

PLOS Pathogens

orcid.org/0000-0002-7699-2064

Thank you for your resubmission. I think that all of the reviewer's comments have now been adequately addressed.
---

## [Editor Report · Acceptance letter]

14 Oct 2024

Dear Dr Akimoto-Tomiyama,

We are delighted to inform you that your manuscript, "BglaTNB6, a tailocin produced by a plant-associated nonpathogenic bacterium, prevents rice seed-borne bacterial diseases," has been formally accepted for publication in PLOS Pathogens.

Best regards,

Michael Malim

Editor-in-Chief

PLOS Pathogens

orcid.org/0000-0002-7699-2064